

# Representing the majority and not the minority: the
# importance of the individual in communicating climate change
Sam Illingworth[1], Alice Bell[2], Stuart Capstick[3], Adam Corner[4], Piers Forster[5], Rosie Leigh[6],
Maria Loroño Leturiondo[1], Catherine Muller[7], Harriett Richardson[8], Emily Shuckburgh[9]
[1]School of Science and the Environment, Manchester Metropolitan University, Manchester,
UK
[2]10:10, London, UK
[3]School of Psychology, Cardiff University, UK
[4]Climate Outreach, Oxford, UK
[5]Priestley International Centre for Climate, University of Leeds, Leeds, UK
[6]National Centre for Earth Observation, University of Leicester, Leicester, UK
[7]Royal Meteorological Society, Reading, UK
[8]National Centre for Atmospheric Science, University of Leeds, Leeds, UK
[9]British Antarctic Survey, Cambridge, UK
*Correspondence to*: Sam Illingworth (s.illingworth@mmu.ac.uk)
## Abstract
This research presents three case studies, through which a creative approach to developing
dialogue around climate change is outlined. By working with three distinct communities and
encouraging them to discuss and write poetry about how climate change affects them, we
demonstrate how such an approach might be adopted at this level. By analysing the
discussions and poetry that arose out of these workshops we show how this community-level
approach to communicating climate change is an essential counterpart to wider-scale
quantitative research. The engagement of each community with climate change is dependent
on the lived experiences of their members; a failure to recognise this results in less effective
communications and can also cause communities to feel isolated and helpless. By considering
the individual needs and aspirations of these communities we can support effective dialogue
around the topic of climate change, and in doing so can better engender positive action
against the negative effects of anthropogenic climate change.

## Keywords

Public Engagement, Climate Change, Dialogue

## 1. Introduction

The communication of climate change has traditionally followed a deficit model (Bickerstaff,
2004), in which a one-way, top-down communication process is adopted. In this approach
scientists have been tasked as the 'experts', whose role is to educate a 'non-expert' general





public, by increasing their knowledge about a particular topic that the experts deemed to be the most significant (Miller, 2001). However, this one-way approach to the communication of climate change is unlikely to bring about the changes that are needed for adaptation and mitigation, as it fails to consider a series of factors that are key determinants of the way people perceive and react to information (Swim et al., 2009). There is not a one-size-fits-all approach that is able to engage society as a whole in regards to climate change. In addition to the type of information individuals need, the way this information is presented will also have an impact on how it is perceived and taken on board. The source of the information is another factor that influences how it is perceived and assessed, and lack of trust in a source, such as the government, the media, or scientists, has proven to affect responsiveness to the message (Goodwin and Dahlstrom, 2014). Information provided by a source that is perceived as untrustworthy and through one-way communication is unlikely to be effective. For example, a lack of trust in the government can affect how people perceive policies in relation to climate change (Lorenzoni et al., 2007).

In contrast to the deficit model, a dialogue model of two-way communication highlights the need to explore the identities and social norms of different groups in society, as well as the importance of acknowledging the existence of many publics - in contrast to what the deficit model referred to as a single public (Priest, 2016). Furthermore, it also acknowledges that the 'non-experts' that constitute the publics also have their own skills and expertise that might also be utilised in the development of research governance (Burns and Gentry, 1998), particularly in the case of these people's own lives and needs, for which they could and should be considered the experts.

The Climate Communication Project aims to understand and evaluate existing expertise in the UK on communicating and engaging the public with climate change. A substantial focus of this project is an expert elicitation (see e.g. de Franca Doria et al., 2009) of the climate communication community, to better understand how a range of specialists carry out their work, to share and promote best practice in the UK, and to point to areas where more investment and attention is needed. This project aims to support and enable a wider structural adjustment to how climate change is discussed and communicated. However, as argued by Lorenzoni et al. (2007) alongside this approach there also needs to be a targeted and tailored information provision to, and communication with, individual citizens and communities. Furthermore, it is essential that the voices of these communities are solicited and considered in the construction of this wider structural adjustment. The work that is presented here reports on a series of dialogues that were established with a small selection of communities across the UK, in order to better demonstrate the importance of these individual voices in developing effective climate change communication strategies.

For this study, a series of three workshops (located in Bristol, Stockport, and Manchester) were coordinated with three distinct and diverse audience groups. Rather than hosting a series of events and expecting members of the community to 'come to us', researchers travelled to established community groups to discuss their needs and potential barriers to considering scientific topics relating to climate change. Three distinct community groups were chosen: the Avonmouth Community Centre in Bristol, Disability Stockport, and a collection of faith groups in Manchester. It is the central thesis of this work that all communities and citizens offer potentially different voices, and as such we did not aim to be representative of



‘every’ community in the UK. Rather we decided to pick a small number of communities in order to demonstrate the value of this approach, and to provide further evidence for its role in developing a more effective communications strategy around climate change.

These three communities were chosen because of their varied composition, and because previous research has highlighted some of the challenges and opportunities of communicating climate change with similar groups. The Avonmouth and Lawrence Western Ward, in which the Avonmouth Community Centre is located, contains areas that are considered to be amongst the most deprived 10% in England (Bristol City Council, 2015). Previous research has shown environmental concerns increase with social class (see e.g. Norton and Leaman, 2004), although actual environmental footprint tends to increase with wealth (Büchs and Schnepf, 2013). Furthermore, since the early days of the environmental movement in the 1960s, community centres have been seen as a potential focus for effective communication strategies (Burgess et al., 1998). By working with the Avonmouth Community Centre we hoped to better understand the role that community centres could play in engaging with people from different social classes.

As noted by Heltberg et al. (2009) the impacts of climate change, even in developed countries such as the UK will sometimes fall disproportionately on vulnerable individuals, with the disabled forming part of the population most at risk from the effects of climate change (Maibach et al., 2010). By working with Disability Stockport, we wanted to ensure that we were giving a voice to the potentially vulnerable, and to better understand their perceptions of how climate change would affect them both as individuals and as a community.

Finally, faith communities tend to share an emphasis on long-term stewardship and can help disseminate information to their publics (Frumkin et al., 2008). By bringing together a group of faith leaders from across Manchester we wanted to get a range of different faith perspectives in relation to climate change, and to better understand how this information was communicated to their respective communities.

As well as the specific opportunities for dialogue in working with each of these communities, it was the aim of this study to demonstrate that these workshops are an effective way of creating a safe space for discussion around climate change. Furthermore, we wanted to show how such an approach could be utilized by other researchers and how this is a necessary accompaniment to any large-scale plans for communicating climate change at a national level or beyond.

## 2.   Materials and Methods

As stated in Section 1, the planned workshops were to take place in the spaces of the selected communities rather than expecting participants to travel to a university or neutral location. The reason for this was so that we could better create a safe space in which participants felt comfortable in discussing how climate change affected their communities, as well as individuals' more general concerns about climate change. In planning these workshops, a two-way dialogue was established between the workshop facilitator (SI) and the community leaders and gatekeepers. Through these dialogues, suitable dates and times for the workshops were decided, with each scheduled to last between two and three hours, and at



times that were seen as compatible with the lifestyles of the community members. Based on
previous experiences and the nature of the activities that were planned for these workshops
(see below), between five and ten participants for each of the workshops was seen as optimal,
thereby ensuring that all opinions could be voiced and discussed in the time allowed. This
number of participants also helped to increase the relative homogeneity within each group
in order to capitalise on people's shared experiences (Kitzinger, 1995) relative to the
community that they were representing.
Following the work of Illingworth and Jack (2018), it was decided that as well as having a
facilitator (SI) and a number of community members, these workshops should also involve
the participation of one climate communications expert. The reasons for this were two-fold.
Firstly, it meant that if any technical questions relating to climate change arose then these
experts would be on hand to provide that information, or else recommend a suitable source
for further inquiry. Secondly, by involving climate communications experts in the workshop,
we hoped to demonstrate to them first-hand the diverse nature of the publics that there were
communicating with. The recruitment of the participants for these workshops was done
through the organisations that we were working with as part of this study, i.e. the Avonmouth
Community Centre, Disability Stockport, and the Manchester Cathedral. Participants were
recruited directly through the community groups and their gatekeepers, with a very basic
flyer provided to each of the organisations so that they could advertise the planned date and
time of the event. Prior to the workshops there were no incentives, financial or otherwise,
offered to the participants to encourage attendance, other than some basic refreshments.
These workshops all adopted a similar format, beginning with a pre-workshop questionnaire
(see Appendix) to be filled out individually by the participants (It should be noted that this
pre-workshop questionnaire actually took place at the beginning of the workshop, prior to
the initial conversations, and so would probably have been better named 'pre-discussion
questionnaire'). This would take place after the initial scope of the research had been
explained by SI and the consent forms had been signed. These responses were to form the
basis of the initial discussions amongst the participants, with their responses acting as an aide
memoire to both help direct the dialogue during the workshops, and also to serve as a record
for data collection. Following this discussion, the participants were guided through a series of
poetry-writing exercises, which involved them first working as individuals and then
collectively to write poetry about two different topics: their community, and climate change.
Poetry was used in this way as it has been shown to be an effective tool in developing dialogue
amongst underserved audiences (Illingworth and Jack, 2018), whilst offering an alternative
form of data collection to complement that recorded in the pre-workshop questionnaire.
These poems were then further discussed amongst the participants, following which a post-
workshop questionnaire was individually completed. This post-workshop questionnaire was
designed to assess the opinions of the participants in relation to the workshop, and to
determine if they had any further questions or required any additional information about
anything that had been discussed. Throughout the workshops, SI made detailed field notes to
later help in the analysis of the responses; this largely took the form of recording and
observing the general nature of the discussions that followed the pre-workshop
questionnaire and the creation of the poetry.

The poetry-writing exercises involved four basic steps:





2. Participants were asked to write a 'list poem' about the chosen topic (either 'your community' or 'climate change'). In this exercise, the participants were given 90 seconds to list everything that they associated with the chosen topic and were reminded that this need not only be things that they could see, but rather that the list could comprise of any associated sense, emotion, or experience.

2. Participants were asked to write one sentence about the chosen topic (either 'How you feel about your community' or 'How you feel about climate change', using the list poem as a word bank for inspiration if required.

3. Participants were then asked to work in pairs and to combine their two sentences. The collaborative effort did not have to rhyme, but it did have to reflect both individuals' observations, and could either be a combination of the two sentences or else something new entirely.

4. Pairs of participants were then asked to work with another pair, and to combine all thoughts and sentences into a coherent piece. Again, this did not have to rhyme, but all participants had to be happy that their thoughts and opinions were reflected in the finished piece.

The poetry writing exercises took place after the initial discussion, as it was hypothesised that this initial dialogue would help the community members to explore their opinions in relation to climate change, both as individuals and as a collective. Furthermore, it was theorised that the poetry would be congruent with these discussions, presenting them in an alternative format that could be shared and analysed alongside the responses to the pre-workshop questionnaire.

All of the questions and prompts that were used throughout the workshops can be found in the Appendix and were also sent to the gatekeepers in advance of the workshops so that their suitability for the participants could be assessed and any necessary provisions to ensure inclusivity could be made. During this study anonymity was preserved by not recording any identifiable information, and during the analysis, any specific or personal narratives that could be seen as identifiable was redacted and destroyed without recording. Furthermore, all the participants were given sufficient time to read the consent forms, so as to avoid assumed consent, and any support workers had access to the consent forms prior to the workshop, so that they could help advise and inform. A suitable line of support was also established through which any distress could be reported and suitably dealt with. By working alongside the support workers all participants knew exactly what the study was for, what it entailed, and what their involvement was. All the support workers were made fully aware of the study, and it was made clear to all participants that they could take part in the activities without having their responses recorded or subsequently analysed. This research project received full ethics approval via Manchester Metropolitan University's Academic Ethics Committee.

3. Case Studies

The findings from the three different workshops are presented as three individual case studies, followed by a discussion in Section 4 about general findings and recommendations in terms of what this approach has taught us. As noted by Moser (2010), more case-specific research is required in relation to communicating climate change, mainly because there is no 'one-size-fits-all solution', with different audiences requiring different narratives, frames,



media and communicators.  By presenting the findings of these workshops as case studies we
hope to better address this requirement, and to also provide further evidence for the need
of this type of qualitative research in order to develop effective climate change
communications strategies.
Each of these case studies will begin with a general overview of the logistics of the workshop,
followed by a presentation of the discussion that occurred following the pre-workshop
questionnaire. The poems that were written by the community groups will then be presented
and contextualized in relation to this discussion, followed by a summary of the key findings
for each community group. With regards to the poems that appear throughout this study,
other than correcting for spelling they are presented exactly as they were written by the
participants during the workshops.
3.1 The Avonmouth Community Centre
This workshop was conducted on a Monday lunchtime, and there were five participants,
including the climate communications expert. The participants were made up of local
residents, volunteers, and people that worked in the area. We spent about 105 minutes
discussing the pre-workshop questions, and about 45 minutes writing poetry and discussing
what this meant and why it had been written.
In the initial discussions around what issues the participants considered to be most pertinent
to their community, better engagement all community members, health (both physical and
mental), and identity seemed to be the most prevalent. In discussing these subjects, the
participants revealed that Avonmouth often felt very geographically isolated ("it doesn't even
feature in some local area maps of Bristol"), and as a result many of the inhabitants found it
difficult to engage with other community groups such as local industries and policymakers.
Furthermore, the issues that people found to be important were acknowledged by them to
be relatively transient, likely to change on a daily basis, and dependent on a range of physical
and psychological factors; for example, litter might be seen as an important issue because
someone threw litter outside their house the previous evening. As well as reporting on being
worried about geographical isolation, the participants also highlighted that this was linked to
their concerns regarding the mental health of their community members, especially the
elderly.
With regards to whether or not climate change affected themselves and their communities
there was initially honest ambivalence, although as one of the participants noted:
39        "I'm not sure people talk about 'climate change' - they may discuss elements such as
40        pollution, seasonal changes / temperatures, recycling, etc."
To corroborate this point of view, when asked to expand on these changes to the climate,
two of the participants (who had lived in the area for the whole of their lives) spent time
discussing how the area was now a lot less polluted than it had been in their youth. With
regards to the pollution of Avonmouth, two of the participants discussed at length how
Avonmouth had once been known for the 'black sheep' caused by the pollution of the
docklands in the 1960s and 1970s. The Clean Air Act of 1970 and its subsequent amendments





(Greenstone, 2004) was likely responsible for the improvement in air quality, although the
participants revealed that to many people "Avonmouth smells". This smell is no longer literal
(and indeed SI noticed no such odour), but this is a view and descriptor that is set in the minds
of many people living in neighbouring districts, thereby possibly contributing to the feelings
of geographical isolation. In 2014, the Environment Agency installed a mobile dust monitor in
the port at Avonmouth, following community concerns about dust (The Environment Agency,
2015). After completing their air quality and dust monitoring work the Environment Agency
were able to demonstrate that air quality in Avonmouth is typical of an urban setting and
should not give rise to an increased risk of respiratory health problems. This monitoring work
was not mentioned by the participants in this workshop but is stated here as further evidence
that the pollution, perceived or otherwise, in this area is something that the community is
deeply affected by. As Bickerstaff (2004) explains, places can suffer 'environmental stigma'
without there being a clear episode of contamination. Stigmatisation can be derived from
perception, and often starts with the very same people who live in that community. Stigma
not only affects the place, but also the people who live in it making them feel trapped, isolated
and powerless. In terms of climate change mitigation and adaptation, stigma is counter-
productive because the feeling of marginalisation and powerlessness can result in inaction or
dismissal of the climate change problem altogether. Therefore, including the views of
communities that feel stigmatised can also be a tool to break this stigma, stop the feeling of
powerlessness, and encourage action.
In discussing what climate change is, and how it may or may not affect the local community,
it quickly became apparent that a perceived conflict within the climate change community
puts people off addressing it, as does the language and negativity that is associated with the
debate centred on this topic. One of the participants stated that:
"People treat climate change deniers like holocaust deniers."
Whilst another participant stated that the way in which climate change is currently
communicated and discussed in the UK:
"Seems like an argument."
These opinions led to a discussion which also revealed that the community members felt that
the politicisation of climate change made it difficult to discuss openly, and as such that it was
almost impossible to "own" and/or take responsibility for. This would seem to advance the
work of Poortinga et al. (2011); i.e. that the acceptance of climate change is not only rooted
in people's core values and worldviews but also what they perceive to be the core values and
worldviews of others. Kahan (2012) has likewise argued that people for the most part take
their cues from peers and own their cultural group on climate change. During the discussion
with community members, it also became clear that the participants were not aware of the
true extent of the consensus amongst climate change scientists, and the majority of them
were surprised when it was revealed that this number was 97-98% (Cook et al., 2016), having
previously believed it to be closer to 50%. The participants also revealed that they were
unclear of where to go for honest and reliable information. Furthermore, some of the
participants considered scientists to be government and industrial stooges, and therefore not



1   necessarily to be trusted. One participant provided further evidence for this opinion in the
2   following statement:

4       "If nutritional scientists are always changing their mind about diet and what is healthy
5       or not, then why should people believe that climate scientists are any different?"

This opinion further supports why one-way communications from such 'experts' will remain
unsuccessful (Lorenzoni et al., 2007). However, by the end of the discussion there was a
general consensus that climate change was something that affected the local area at both the
community and the individual level, and that in order to better relay this information and
discuss what could be done to mitigate its effects, there was a need to move away from a
'one-way forum' and towards a 'conversation café' i.e. the creation of an environment in
which these conversations could take place in a shared space and where no one would be
judged. Conversations then turned towards what difference a single individual could make,
and if asking this question was having a negative effect on discussing climate change and
whether or not people could realistically be expected to take on this personal
responsibility. This discussion featured input from SI and the other expert in terms of
answering technical questions and providing information such as the true figures for
consensus amongst scientists studying climate change. However, neither SI nor the expert
acted in any way so as to persuade or dissuade any of the participants from a particular way
of thinking.
Following these discussions, the following two poems were written collectively by the local
community participants. On the subject of 'How you feel about your local community':
Looking back through today's eye at
an interesting, friendly place full of history
appreciating what we have
a bit dishevelled, sometimes unloved
but with potential to thrive
feels caring, friendly, home
loving where we live and work.
And on the subject of 'How you feel about climate change':
Confused, conflicted, guilty, sad, helpless but I have a
responsibility to educate myself, live simply and do whatever
I can to affect positive change… we can educate people
to the real statistics of what is happening in our world.
In discussing these two poems, the participants made it clear that for both subjects (i.e. their
community and climate change) whilst work was needed to improve the current situation,
hope was not lost. In reading these poems, it is clear that the participants have a strong sense
of civic pride in their local community, and that it is a place that they are genuinely proud to
call home. Furthermore, they believe that they have a duty of care to improve their
community and the lives of those people in it, and that this extends to the effects of climate
change. Given the lengthy discussion on the consensus of climate change scientists and the





surrounding ideas of media bias, it is unsurprising that it features so prominently. On reading these poems it is also evident that the participants believe they have a responsibility to affect positive change and to educate people. The collective poem on climate change that they wrote accurately summarised the previous discussion (even though this was not explicitly or implicitly expressed to the participants prior to the exercise), i.e. that there was a desire to have an open and honest conversation in a safe environment, and that this approach could then be used to educate others so that they could also make up their own minds. It should be noted that throughout this study, there is no emphasis placed on the aesthetic quality of the poetry, and that by emphasising this to the participants it was easier to create a shared space for creativity and sharing.

From the post-workshop questionnaire, the main issues that people still wanted to address were what they could do to help, whether they were too late to help, and where the best resources were to find out more about climate change and how to mitigate its effects. Overall everyone seemed to enjoy the workshop, although they would have liked even more time to work on their poems. A response of note for this section of the questionnaire was that one of the participants now felt as though they would come to the workshop facilitator (SI) for more information about climate change; previously this participant had been sceptical of trusting scientists for the reasons outlined above. Furthermore, this participant contacted SI a couple of weeks after the workshop with the following request:

> "I have been thinking a lot about the workshop and I was wondering if it would be ok to use the idea of it with other people. I wanted to try doing it with the Quaker children meeting and our lunch group."

This request serves to underline the effectiveness of the approach that was adopted for this workshop; by creating a safe space in which dialogue could be established and individual voices could be heard and listened to, the perceptions of scientists changed from untrustworthy to valued and reliable sources of information; in this case with the added advantage that the approach was adopted and taken on in another context. This workshop also highlighted the potential roles that community centres can play in providing a safe space for discussions surrounding climate change in a neutral and non-politicised environment. Shortly before the workshop in Avonmouth, SI also spoke to a group of 'Community Payback' young men who were having their lunch in the community centre. In these conversations, they were respectful and honest in informing SI that they did not care at all about climate change, and that there was no point as "the world was going to end anyway". They were perfectly happy to talk to SI and to express these views but did not want to engage further on the subject. Perhaps it is the community volunteers of Avonmouth who are better served to engage this audience around the effects of climate change, and to help demonstrate how despite being "a bit dishevelled, sometimes unloved" they have "potential to thrive". The effectiveness of involving mediators who already have access to harder-to-reach communities, who are already trusted by these communities, and who understand the community's ecology is also highlighted in other studies with a similar purpose (e.g. Ramírez et al., 2015).

3.2 Disability Stockport




This workshop was conducted on a Monday afternoon at Disability Stockport, with five participants, including the climate communications expert. The participants were made up of volunteers and patrons of Disability Stockport, including one participant with severe learning difficulties who needed support throughout the workshop. This support was provided by SI who worked with this participant on a one-to-one basis, and then helped to feed back their input to the rest of the group during the discussions and poetry-writing exercises. We spent about 80 minutes discussing the pre-workshop questions and about 40 minutes writing poetry and discussing what this meant and why it had been written.

In the initial discussions about what the participants found to be important in their local community, social justice and equality for all were the dominant topic of conversation. The participants were finely attuned to inclusivity and wanted to ensure that all of their community members had a strong and discernible voice on matters that affected them, even if they were not necessarily aware that this was the case. In talking to the more vulnerable participants and their carers, it became apparent that they are completely reliant on friends and family members for information on most topics, and so it is vital that these people are equipped with the correct information and tools to help further engender this communication. Any biases, perceived or otherwise, that these carers and volunteers are subjected to will likewise be passed on to the vulnerable members of the community that they help to support. In discussing the issues that were most important to the local community, the importance of living in a healthy environment was raised repeatedly, and what this meant in terms of both physical and mental wellbeing. As with the Avonmouth community, the mental health of the community members, and the risk of isolation and exclusion that this could bring, were also seen as very important issues.

With regards to climate change, the responses from the participants were varied. The volunteers appeared to be very aware of the subject and how it affected both them personally and also the people that they cared for and the wider community. This is perhaps reflective of the several sustainability initiatives that Disability Stockport has led and been involved with, including its use of compostable recycling and the installation of solar panels on the roof of their building, which they self-funded through fundraising events (Crush and Cameron, 2015). However, the more vulnerable members of the community were much less aware about climate change and the effects that it would have on them. This awareness ranged from a feeling that climate change was 'bad' but an inability to articulate why this was the case, to having absolutely no concept of the processes or effects of climate change. This lack of awareness as to the existence of climate change might in part be explained by the way in which it is communicated, with one of the volunteers stating that this was done by:

> "the usual suspects… through interest groups like F.O.E., the UN, The Guardian, and Greenpeace."

The participants felt that as well as the 'usual suspects' attempting to communicate climate change, the audience that they were communicating to also consisted of the 'usual suspects' and did not tend to include the members of their community, both in terms of Disability Stockport and Stockport more generally. However, as one of the participants pointed out:

> "These people represent the majority, not the minority."



In order to better engage this majority, participants believed that climate change communication activities needed to happen at other more 'regular' events. A local example of a 'hate crime' awareness event that had a band and other activities and was not advertised as a 'hate crime awareness event' was discussed as a good model, as it had attracted a large cohort and generated effective and meaningful discussion. According to one of the volunteers, Stockport used to have a very good local environment fair that did communicate issues relating to sustainability and environmental change, in an accessible manner and to a wide audience; this fair was allegedly very popular, but austerity and local government cuts meant that it was cancelled. This failure of the local and central government was a topic that was repeatedly brought up in this workshop, and there was a strong belief that there was a need for policymakers and government to shoulder the majority of the blame for the negative effects of climate change; as one participant put it:

> "When will our social leaders agree to effect change and find ways to overcome collective greed?"

Stockport is part of Greater Manchester, and Devolution to the Greater Manchester Combined Authority (Copus et al., 2017) was seen by the participants as a great opportunity for enacting positive change in terms of both equal rights and mitigating climate change. The approach that was adopted by Ken Livingstone whilst he was the Mayor of London (2000 – 2008) was stated as a good standard to follow (Shove and Walker, 2010), and the participants hoped that Andy Burnham (the first Mayor of Greater Manchester) would use his newfound responsibilities and power in a similar fashion. This discussion featured input from SI and the other expert in terms of answering technical questions. However, neither SI nor the expert acted in any way so as to persuade or dissuade any of the participants from a particular way of thinking.

Following these initial discussions, two poems were written collectively by the participants. On the subject of 'How you feel about your local community':

> I think community is being lost, everyone's too busy.
> I feel close to my community and part of it.
> I feel like there are many selfish people
> But there are people who help.
> My community is a lonely concrete desert where desert flowers bloom,
> sometimes,
> if they catch a bit of warm rain.

And on the subject of 'How you feel about climate change':

> Some will profit as suffering increases.
> Sorry, sorry, sorry, sorry, sorry children of the future!
> We have one Earth, if we don't save it, all else is lost.
> I feel like if I give as hard as I could
> My friends will live in a world that's good.





In discussing these two poems, the participants again returned to themes of social justice and what was and was not perceived to be 'fair'. They found it grossly unfair that a minority of people were spoiling both their community and the local and wider environments for the majority. They also discussed how despite this selfish minority, there were other people who were acting as a force for good, and who could, and should, be relied upon to help enact a positive change. As was the case with the Avonmouth poetry, both of these poems were reflective of the previous discussions (although it was perhaps surprising that local and national authorities, and their perceived failings in terms of austerity and sustainability, were not explicitly mentioned). In particular, the last two lines of the collective poem about climate change effectively summarised the prevailing mood of the group, which was ultimately one of hope and empowerment. Rather than a burden that caused them to feel belittled and helpless, the volunteers in the group saw it as an opportunity to provide the support that was needed to help the unaware and the vulnerable, both within their own community and beyond. As with the previous discussion, it became apparent that this community was comprised of two distinct groups of people: the volunteers and carers, and the people that they helped. Whilst certain circumstances dictated that some of the participants spent time in both of these groups, the poetry that was created and the subsequent discussions made it clear that any climate change communication strategy that aimed to effectively work with this community must target both of these publics.

Given the restrictions that Disability Stockport, and other communities like them, have faced because of funding cuts brought about by austerity measure in the UK (see e.g. Cross, 2013), it is perhaps unsurprising that the volunteers within this community are aware of the responsibilities of both local and national government, and that they are willing to take them to task on the matter. In contrast to the participants at the Avonmouth workshop they did not express a restraining sense of guilt, but rather an acceptance that they could not, and should not, be held individually responsible for the effects of climate change and our attempts to mitigate these changes. This community is very firmly attuned to a sense of justice, and they want to ensure that everyone has a strong and discernible voice in discussing climate change, not least because they recognise that whilst many of their members are contributing the least to climate change, they will be amongst the ones that are most affected by it.

From the post-workshop questionnaire, the main questions that participants still had were related to how they could help others (especially locally policymakers) to take collective responsibility for their actions. The participants appreciated the "egalitarian, respectful, and non-judgmental" creative approach to the workshop, and its success in "including disabled people fully." One request that was made was for links to local groups and information relating to the communication of climate change to be made available, which further corroborates the desire of the participants to help others take notice and "motivate those in charge".

This workshop demonstrated how important it is to fully consider the vulnerable members of our society when thinking about how climate change and its effects are communicated. As well as ensuring that any communication strategy is not just aimed at the 'usual suspects' it is essential that the carers are also well equipped with the tools and information to help engender meaningful and unbiased debate on the subject. Furthermore, by giving these communities a voice, any efforts to communicate the effects of climate change would stand



### 3.3 Manchester Faith Communities

This workshop was conducted on a Thursday afternoon, and there were eight participants, including the climate communications expert. The workshop took place in the refectory of the Manchester Cathedral, with representatives from the Catholic Church, Protestantism, Judaism, and the Bahá'í Faith. Each of these representatives were leaders within their faith organisations and the initial discussions lasted approximately 80 minutes, with 60 minutes spent collaboratively writing and discussing poetry.

Initial discussions with this group focussed on what was meant by the word 'community', with participants discussing which communities they did and did not belong to. For the faith leaders that were represented here, they all felt part of their faith communities, but also the local communities where they lived, as well as more regional, national, and even global non-faith communities. This attitude of belonging to a global community was summed up by one participant:

> "We all belong to the wider community of humanity. We all bleed red blood, we all breathe the same air."

With regards to issues that were seen as pertinent to their local faith communities, the environment and food awareness (i.e. food waste and food poverty) were highlighted and discussed at length. All of the participants felt that these issues could be addressed in a meaningful and effective manner by first better developing educational awareness around these topics, and by promoting better interconnectedness, both between the communities and across the topics of importance. As with the other two workshops, the importance of a healthy environment was discussed at length, and all of the participants expressed (without being prompted) that the effects of climate change were amongst the greatest issues that they were currently tackling in both their local and wider faith communities.

This was a very informed group in terms of climate change and its effect on both individuals and their wider communities. Given that this workshop was advertised as an opportunity to discuss climate change, this might be expected, but as was revealed in the discussions, many of the faith communities are already taking considerable steps to address the effects of climate change at both a global and a more local level. Organisations and initiatives such as Green Bishops (Dakin, 2004), the Public Issues Team at Methodist Church House (The Methodist Church, 2012), and Pope Francis' *Laudato si* (Francisco, 2015) were all discussed as both sources of inspiration and useful references for further information. From these discussions it was apparent how each of these faith leaders belonged to a much larger community that they could work with and on behalf of, and as with the volunteers within the Disability Stockport community, these participants believed they had a duty of care to help improve the environments of the more vulnerable members of their communities. There was also an extended discussion about how many of the more vulnerable members of these communities were seen as "problems that needed to be solved", whereas they should instead





be viewed as potential solutions to many of the issues facing the communities, especially those surrounding the effects of climate change. As one of the participants noted:

"If people knew then they could make any informed decision."

Despite their own knowledge on the subject of climate change, and the resources that were available to them through their faith communities, the participants still expressed a need for reliable and unbiased information that they could then direct their communities to. All of the participants believed that whilst the effects of climate change were going to have a negative effect at both a global and local level, these challenges also presented an opportunity to bring people together and empower the impoverished by working in unison to tackle the negative effects of climate change. This discussion featured very little input from SI and the other expert in terms of answering technical questions, and nobody acted in any way so as to persuade or dissuade any of the participants from a particular way of thinking.

Following the initial discussion, the participants were split into two groups of four, and worked in these groups to create two sets of poems. Two on the subject of 'How you feel about your local community':

Community is the space where we
are cherished and appreciated, a place
of encounter where all belong,
Supporting each other with a
common vision; we are a kaleidoscope of life.

And

I like my community - its resourceful people with familiar sparkling eyes of hope,
sensing potential to beautify.
Strangers need not feel alone
Where diverse community cherishes home.

And two on the subject of 'How you feel about climate change':

I have come to see that climate change affects us all
My consumption is at the expense of my neighbour's lack
And my recklessness may lead to my neighbour's danger
My careless lifestyle causing so much natural beauty to be lost
I sense the urgency that I change to help save the planet
For the future me that this haunts drives me, transfuses my life.

And

There are too many of us
Disposing of too much fare
Into our atmosphere and our world
We need to take more care,





| | |
|---|---|
| 1 | Fossil industrial growth |
| 2 | That diminishes water soil and air |
| 3 | Grow to green and clean |
| 4 | To make the world more fair. |
| 5 | We need to change behaviour |
| 6 | It is urgent that we share, |
| 7 | The joy is living simply |
| 8 | Right here and not out there. |
| 9 | We must reduce the harm we cause |
| 10 | Both personal and corporate ware |
| 11 | A better carbon footprint |
| 12 | Before our world we tear. |
| 13 | |

These poems, and the discussions that followed, served to further highlight the congruence
between these participants. Unlike the participants in the Stockport and Avonmouth
workshops, this group did not all belong to one common community, but the similarities in
their beliefs with regards to their collective responsibility was striking. From these poems it is
clear that the faith leaders consider communities to be places of strength and belonging, and
that we should work hard to connect these communities so that nobody is ostracised; it is the
similarities between communities rather than their differences that should be cherished and
nurtured. These participants accepted their collective guilt with regards to the effects of
climate change, but also saw it as an opportunity to develop cohesion and belonging amongst
the most vulnerable. As with the Stockport group, they realised that they had a responsibility,
but saw this as something that was achievable rather than overbearing.
Both of the poems written about climate change recognise that the negative consequences
to climate change (and any response to it) have come about because of an imbalance. The
line "My consumption is at the expense of my neighbour's lack" is very similar to the ideas
that were expressed by the Stockport group, i.e. that the privileged minority has been living
at the expense of the disadvantaged majority, and in many instances has been responsible
for maintaining and even strengthening that disparity. On reading the lines "There are too
many of us / Disposing of too much fare", Thomas Malthus and the relationship between
population growth and climate change might initially spring to mind (Kelly and Kolstad, 2001).
However, these lines should also be read alongside "The joy is living simply / Right here and
not out there". It is not necessarily rapid reductions in population growth that are being
advocated in this poem, but rather the notion that we need to better consider exactly what
is meant by 'sustainable living' and the changes to our personal lifestyles that might be
necessary in order to mitigate the negative effects of climate change for everyone (Carley and
Spapens, 2017). These poems do not promise easy answers, and they also point to a sense of
immediacy, i.e. that something needs to be done now, and by the authors of these poems,
rather than waiting and hoping for a future solution or future author to present itself.
As with the Avonmouth group, these poems (and the surrounding discussions) pointed to a
need for open and honest debate, and with it an interconnected approach to educating
people in a safe environment; one in which they felt welcome and cherished. Throughout all
of the discussions there was a willingness to assume collective responsibility, and a desire
amongst the participants to use their positions of responsibility to not only help their



communities, but to work together so that they might better tackle the negative effects of climate change. As one of the participants noted:

"It is about overcoming prejudices."

This comment was made in relation to how different faith communities could more effectively work together, but it is also relevant in regards to the need to go beyond the 'usual suspects' when determining the audiences and the associated messages for the effective communication of climate change.

From the post-workshop questionnaire, the response of the participants was similar to that of the Stockport group, as they mainly wanted to know more information about "how to inspire more behaviour change and faith-based action", with both groups explicitly wanting to know how they could "activate hope". The participants enjoyed the creative elements of the workshop and liked the "focus on participation" and the "fun and accepting" nature that accompanied the "serious discussion". As with the Stockport group, they would have liked some practical examples of what they could do to enact change, both within their faith communities and beyond.

This workshop succeeded in bringing together a group of faith leaders from across Manchester, to present a range of different faith perspectives in relation to climate change. These are strong and interconnected communities that want what is best for all of their members, but not at the expense of other more vulnerable members of society that might not belong to their community. The participants in this workshop represented a well-informed and powerful agent with regards to the effective dissemination and communication of climate change and working with these faith leaders to develop dialogue within and across their communities is something that should be better considered by climate communication strategies.

## 4. Discussion

In reading these case studies, and by analysing the discussions and the poetry that were generated in the workshops, it is evident that each of the three communities has a clear and distinctive voice. These distinct voices mean that there are distinct challenges in effectively developing dialogue around climate change, but as can be seen from Section 3, there are also diverse opportunities in working *with* each of these communities to better develop this dialogue.

In all three of the communities there was a sense of collective guilt, centred on a recognition of personal responsibility; that we as individuals were at least partly to blame for the negative effects of climate change that were observed at both an individual and community level. However, how each of those communities reacted to notions of personal and community responsibility was distinct and serves to highlight why a 'one-size-fits-all' approach to communicating climate change, or even developing dialogue around the subject, would not work. The participants in the Avonmouth workshop initially largely felt overwhelmed and de-motivated by their guilt. So much had already gone wrong how could they as individuals now help to set things right; it seemed like potentially an overwhelming task, and they felt





"Confused, conflicted, guilty, sad, helpless". But through discussions amongst themselves and a sharing of that guilt they came to the realisation that they "have a / responsibility to educate myself, live simply and do whatever / I can to affect positive change". In order for a community like the Avonmouth Community Centre to enact positive change, they need to be freed from any individual guilt, which itself has maybe been deepened by previous (one-way) climate change communication efforts.

In contrast to the Avonmouth group, whilst the Stockport group also acknowledged their guilt, they recognised that they were not solely responsible for the current negative effects of climate change. Furthermore, they recognised that through their actions they could make a positive difference: "I feel like if I give as hard as I could / My friends will live in a world that's good." Contrast this to the "we can educate people / to the real statistics of what is happening in our world" of the Avonmouth poem. There is a greater degree of certainty (still not absolute) that they can enact positive change, both as individuals and as a collective. In working with a community like Disability Stockport, effective communications would likely highlight ways in which others (e.g. governments and policymakers) could be held to account for their collective failings.

The community of faith leaders had a similar outlook to the Stockport group, recognising that: "We must reduce the harm we cause / Both personal and corporate ware / A better carbon footprint / Before our world we tear." And that "To make the world more fair. / We need to change behaviour". As with the Avonmouth group, they also realised the need for education, and given their own positions within their communities they recognised that any initial activity likely needed to be driven by them. This was arguably a different type of individual responsibility than was evidenced in the other two workshops, as the faith leaders recognised that in some instances without their guidance and support for a particular topic action might not be instigated or even possible. In working with this community, it could be argued that effective climate change communications would provide reliable resources and frameworks for engagement that could then be shared by the individuals amongst their own communities and organisations.

The manner in which guilt about climate change was attributed, and the extent to which it oppressed individual and collective action, is just one example of the different ways in which these communities responded to climate change and how it is communicated. People's individual roles within these communities also need to be considered. For example, are they resident or employee; volunteer or patron; faith leader or community member? These roles may change depending on circumstance, and many of us belong to several communities, in which we might have different roles and react accordingly. Given these different communities and the roles within them, how do we go about categorising them in terms of developing effective climate communications? Helm et al. (2018) have suggested using an approach that splits people's values into egotistic, altruistic, and biospheric, but is even this approach too broad? As noted by one of the participants in the Manchester workshop:

> "Different people respond to different stimuli. Express themselves very differently, so
> how to engage will vary according to the audience / psychological makeup of hopes
> and fears."





By making generalisations about how to effectively communicate climate change we are
missing these reactions, and in doing so we are arguably contributing to a perceived malaise
on the subject. Furthermore, but not working at the community level we are missing out on
all of the opportunities that these communities (and their individuals) present in terms of
developing effective dialogue around the negative effects of climate change and mobilising
collective action against them. Whatever the theoretical perspectives on how people's
opinions and values can be categorised, they are typically unable to recognise the very
particular circumstances that are present in individual communities. Nevertheless, each of
the three communities in this study represent effective allies towards the mitigation of
climate change. The Avonmouth Community Centre were willing to engage their own
member base and wanted to depoliticise climate change so that they could educate their
community how best to combat its negative effects. Disability Stockport understood the
social injustice of climate change and were willing to bring to task local government in order
to protect the vulnerable. The Manchester faith leaders were eager to use their positions
within their own communities to educate, support, and enact change. These are all positive
experiences and opportunities, which serve to highlight the question of why we are not
working with these communities instead of telling them what they should be doing and how
they should be feeling.
The approach that was adopted in this study has helped to give voice to a small selection of
different communities, and in doing so has helped us to better understand why there is no
'one-size-fits-all' approach to communicating climate change. It also highlighted why two-way
dialogues are needed to help capture and understand these approaches, as opposed to one-
way communications which can instead instil negative feelings and attitudes. By creating a
safe space in which dialogue could take place, these workshops helped to empower the
community members, and in using poetry as part of the process the participants were
presented with a creative approach to solidify their thoughts and communicate and discuss
them with others. The poetry also acted as a powerful tool in helping participants to explore
the lifeworlds of their associates and enabled them to reflect on what had been discussed
and what they might decide to do in the future. Whilst poetry can at times be perceived as
elitist and 'difficult', these workshops demonstrated that given the correct environment and
facilitation, writing poetry can instead be accessible and empowering. None of the workshops
participants had any issues in composing their poems, and indeed almost all of them took
great joy in creating and sharing them.
The creative nature of these workshops was enjoyed by all of the participants and
demonstrates how poetry can play a powerful role in helping to develop effective dialogue
around climate change. During the workshops, several of the participants noted that this kind
of activity should be run elsewhere and that it was needed to help ensure that all voices could
be heard. Based on these experiences the following recommendations are offered to people
wanting to adopt a similar approach:
1. These workshops need to happen in the communities themselves. It is not desirable
(both in terms of logistics and the creation of a safe space) for these workshops to
happen at a university or even a neutral venue;





2. Any workshop questions or planned exercises should be passed to a community representative or gatekeeper in advance of the workshop, so that provisions can be made to be fully inclusive;

3. In order for everyone to be equally involved in the discussions an upper limit of 10 people, or 10 people per facilitator, would be advisable;

4. The role of the facilitator is not to be overlooked. This needs to be someone who can respond to questions, support groups discussions, assist in poetry writing, and quickly synthesise information. Several facilitators, each with a slightly different specialism (e.g. poetry writing and group discussions) might be advisable;

5. Having regular breaks, and creating an informal atmosphere helps to breed creativity and also reinforce the notion of a safe space for all.

As discussed in Section 1, we hoped that by involving climate communications experts in the workshop, we could demonstrate first-hand to them the diverse nature of the audiences and publics that there were communicating with. In conversations with the experts following these workshops this was clearly the case; in all instances it was useful to have someone who could not only provide statistics and in-depth information if required to do so, but who could also offer an alternative opinion and voice in terms of their own communities. In future workshops it might also be worthwhile to include a climate communications expert who identified as also being part of the community group that is being worked with, so as also to provide local information and an additional representative voice.

This study is limited in its findings, in that we only report on the outcomes of three workshops run in three different community groups. The findings would likely be very different were these workshops to be run again but with different communities. However, this further serves to underline the thesis of this study, i.e. that qualitative research at the community level is an essential accompaniment to larger scale research projects that look at the way in which climate change is communicated. One-off workshops were used in this study, as we believe that it represents a model that could be most easily adopted by other researchers and for other communities. Additionally, this study was not designed to monitor the long-term impacts of these workshops; however, given the responses of the participants (and in particular the comments made by the Avonmouth group – see Section 3.1), such a study would likely yield interesting results. In addition to working with different communities and monitoring any long-term impacts, future studies could also adopt a similar approach to running workshops with several communities at a time. As demonstrated in this study, the collaborative poetry writing worked well in allowing participants to explore each other's lived experiences in a creative and non-confrontational manner. Such an approach would also likely be successful in helping to bring together different (and perhaps opposed) communities by enabling them to discuss their lifeworlds in this way, as was exemplified by workshop involving the Manchester faith leaders (see Section 3.3).

5. Conclusion

This study has presented a framework for engaging communities in an effective dialogue around the effects of climate change. In presenting the results of these discussions via three case studies, we have also highlighted the need for such initiatives, both in terms of better understanding the needs of these communities, and also the opportunities that they present





in mobilising effective action against the negative effects of climate change. In addition to the specific needs and opportunities for each of these communities, this study has also demonstrated how poetry can help community members to explore their own and each other's lifeworlds in a creative environment, and in doing so has shown how workshops such as these are an effective way of creating a safe space for discussion around climate change.

This approach has also provided evidence for how a dialogue model can help to break down some of the barriers that are created via one-way communication exercises. By creating a safe space in which dialogue could be established and individual voices could be heard and listened to, the perceptions of 'experts' changed from untrustworthy to valued and reliable sources of information. In developing this dialogue, it is vital to also realise the different roles that individuals play within different communities, and when working with carers and other gatekeepers a consideration needs to be given to how they too can be supported in developing their own effective dialogues.

The three communities in this study represent only a small fraction of the different audiences and publics that need to be engaged with, in order to effectively develop a dialogue around communicating climate change and bringing about the changes that are needed for mitigation against its negative effects. The small-scale, creative, and personal qualitative research that is presented here is essential to help contextualise and develop larger impersonal quantitative work, demonstrating that whilst we are multitudes we are also individuals, and that all voices should be listened to and taken into account. Such engagement should not simply be done as a box-ticking exercise but should be encouraged because diversity and inclusion acts as a powerful tool for empowering citizens and enacting change (see e.g. Stevens et al., 2008). By telling individuals what they can and cannot do, and how they can and cannot feel in relation to climate change, we are arguably contributing to a feeling of collective guilt that can entrench feelings of defensiveness and despair. By listening and giving voice to each of these communities we can not only help to break down these barriers, but in doing so can benefit from their unique skill sets and experiences as future allies in our battle against anthropogenic climate change.

Acknowledgements

The authors would like to thank and acknowledge all the participants in this study, including the staff at Avonmouth Community Centre, Disability Stockport, and the Manchester Cathedral for their help in making this project possible.

This work was supported by the Natural Environment Research Council (NE/R011974/1).

Author contributions

SI designed and delivered the workshops, analysed the responses, and co-wrote the paper.

AB, SC, AC, PF, RL, MLL, CM, HR, and ES helped design the workshops, analyse the responses, and co-wrote the paper.





Competing financial interests
The authors declare that there are no competing financial interests.

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

E. 2009. Psychology and global climate change: Addressing a multi-faceted
phenomenon and set of challenges. A report by the American Psychological
Association's task force on the interface between psychology and global climate
change. *American Psychological Association, Washington*.
THE ENVIRONMENT AGENCY 2015. Avonmouth: dust monitoring. The Environment Agency
THE METHODIST CHURCH 2012. Hope in God's Future: Christian Discipleship in the Context
of Climate Change. London, UK: Methodist Publishing.
47





Appendix
*There is no demographic information on this questionnaire for two reasons. Firstly, it assures*
*that the responses are fully anonymised. Secondly, we are interested at communicating with*
*people as people, and as such generalisations relating to gender, race, age, and any other*
*socio-demographic factors should be discouraged.*
**Pre-Workshop Questions**
Write down three random words. This question is needed to help analyse the responses.
What are the three most important issues that need addressing in your community?
Does climate change affect your community?
Does climate change affect you?
What is climate change?
How do you think climate change is currently communicated?
What do you want to know more about with respect to climate change?
How would you find out this information?
**Workshop Questions**
Write a list poem about the things in your community.
Write down one sentence that captures how you feel about your community.
Combine this sentence with a neighbour.
Combine this pair of sentences with another pair
Write a list poem about climate change.
Write down one sentence that captures how you feel about climate change.
Combine this sentence with a neighbour.
Combine this pair of sentences with another pair.
Write down one question that you have about climate change.





**Post-Workshop Questions**
What did you like about this workshop?
What could we have done differently?
What is climate change?
What do you want to know more about with respect to climate change?
How would you find out this information?
