# Peer review of "Representing the majority and not the minority: the 1 importance of the individual in communicating climate change 2 3 Sam Illingworth1, Alice Bell2, Stuart Capstick3, Adam Corner4, Piers Forster5, Rosie Leigh6, 4 5 Maria Loroño Let"

_Geoscience Communication, 2018_

## Referee Comment (RC1) · E. Miller (Referee) · 12 Jun 2018

Overall, this was a very innovative and interesting paper. I really enjoyed the method and analysis, and what we can learn from this type of research / community engagement approach. That said, a paragraph explaining the value, impact and process of research poetry as a both a tool of engagement, advocacy and communication would strengthen the paper, as would some reflections on why poetry.

---

## Author Comment (AC1) · 14 Jun 2018

Thank you for your kind comments. We agree that a paragraph on the potential value and impact of research poetry would be of great benefit to the reader and would also strengthen the paper. We have constructed the following paragraph to address this issue, which also reflects on why poetry (rather than another artistic medium) was used in this approach; this paragraph will be inserted in Section 2 (Materials and Methods), where the poetry-writing exercises are first introduced.

Poetry can used to help reframe and develop dialogue amongst participants and has an established history as a tool that can be used by researchers to both communicate with

and elicit engagement amongst different audiences. For example, by turning participant recordings and transcripts into poetic performances, Finley (2003) demonstrated how poetic responses might be used to open up new dialogues with communities, using their own words but presented in an alternative format. Similarly, poetry that is written by participants can be used as data by researchers to better understand the lifeworlds of the authors, serving as powerful narrative examples in the development of education and advocacy goals (Poindexter, 2002). By asking the participants to write their own poetry, we hoped to enable them to consider their thoughts and opinions in a creative space, which could then be analysed alongside their non-poetic responses. The reasons that poetry was used rather than another artistic medium (e.g. sculpture or drawing) were two-fold. Firstly, the workshop facilitator (SI) has experience in both creating poetry and running poetry-writing workshops, as such he was able to play the role of what Vygotsky (1980) termed the 'More Knowledgeable Other', and in doing so could help to extend the social learning of the participants. Secondly, poetry writing is a very accessible activity that only requires paper and pens / pencils, and which can be both easily transported and also supported; for example, with regards to participants who are themselves unable to write.

References

FINLEY, M. 2003. Fugue of the street rat: Writing research poetry. International Journal of Qualitative Studies in Education, 16, 603-604.

POINDEXTER, C. C. 2002. Research as poetry: A couple experiences HIV. Qualitative Inquiry, 8, 707-714.

VYGOTSKY, L. S. 1980. Mind in society: The development of higher psychological processes, Harvard university press.

---

## Referee Comment (RC2) · Anonymous Referee #2 · 30 Jun 2018

General Comments: This paper will be of interest to researchers across different disciplines, particularly those who are considering doing outreach or public engagement of their own. I found its justifications, methods and materials to be clear and coherent and it raises some important points about the use of qualitative research embedded in communities.

Specific Comments:

A couple of small ones.

Poetry: I agree with the first reviewer's comments that the reasons for selecting poetry needed to be given slightly more space. The authors' proposed additional paragraph

goes much of the way towards rectifying this. It strikes me that many of the positive points here could be applied to other types of creative-writing exercise, and it seems that one of the reasons was the expertise of the workshop co-ordinator as playing the role of 'more knowledgeable other'. (No Problem with that.) However, at certain points the article touches on some poetry-specific features e.g. the way participants really engaged with poetry despite it being seen as elitist/difficult'. There are also some considerations of the way these workshops could be re-prised/repeated elsewhere. I think it is worth having a couple of sentences at least considering, during future/follow-up workshops, poetry could be more than, as the authors call it, a 'tool'. After all poetry has a highly developed (and comparatively accessible) tradition of thinking about and engaging with both place/community and 'nature'. In sum: Is it worth saying something about whether such community workshops could incorporate the reading as well as the writing of poetry, even if only to rule it out?

Religion: Hearing how the participants from different faith communities engaged with the workshop was one of the most interesting parts of this article. I agree with the conclusion that working with faith leaders to develop dialogue across the diverse communities is a worthwhile initiative, and that awareness of different publics' perspectives, needs and worldviews is part of climate change communication. Therefore it strikes me that the article could do slightly more to engage with the relationship between religious discourses and ecological awareness in their own terms rather than too quickly putting them into an already familiar language of sustainability. The things in the discussion about neighbourly responsibility, or living in the moment with less focus on consumption, are not just connected to community experience but in part emerge from a religious world view that might complement but also find itself in tension with aspects of ecological communication. And while the implications need not be discussed in detail here they could perhaps be better acknowledged/signposted in a few sentences (e.g. in regard to Laudato Si, or even a Quaker sense of stewardship etc etc.).

Both of the above are really quite small quibbles and would only require very minor

amendments.

Technical Corrections: None.
* * *

---

## Author Comment (AC2) · 2 Jul 2018

Thank you for your kind words, and for your useful critique of this research. We will now respond to the two specific points that you have mentioned in turn.

Poetry: I agree with the first reviewer's comments that the reasons for selecting poetry needed to be given slightly more space. The authors' proposed additional paragraph goes much of the way towards rectifying this. It strikes me that many of the positive points here could be applied to other types of creative-writing exercise, and it seems that one of the reasons was the expertise of the workshop co-ordinator as playing the role of 'more knowledgeable other'. (No Problem with that.) However, at certain points

the article touches on some poetry-specific features e.g. the way participants really engaged with poetry despite it being seen as elitist/difficult'. There are also some considerations of the way these workshops could be re-prised/repeated elsewhere. I think it is worth having a couple of sentences at least considering, during future/followup workshops, poetry could be more than, as the authors call it, a 'tool'. After all poetry has a highly developed (and comparatively accessible) tradition of thinking about and engaging with both place/community and 'nature'. In sum: Is it worth saying something about whether such community workshops could incorporate the reading as well as the writing of poetry, even if only to rule it out?

This is an excellent point, and we agree that poetry can (and should) definitely be used in this manner. When we spoke about poetry being used as a 'tool', we wanted to make the distinction between it being used as a 'tool' and being used as 'data' to be analysed and considered. However, it is absolutely necessary that we highlight how reading (and even analysing) poetry might be used in such community workshops to great effect. As such, the paragraph that was added in response to Reviewer 1 was addended with the following:

It should also be noted that reading and analysing (as well as writing) poetry can also be used to engage different audiences with specific topics, and that there is a history of such initiatives being used to successfully explore different relationships and opinions across and between communities (see e.g. Furman et al., 2004). However, for the purposes of this research, we chose to focus on writing poetry as it allowed for the most collaborative experience within the framework of the workshops.

Furthermore, the Final paragraph of Section 4 ('Discussion') was also amended to reflect how reading poetry might be used in future workshops:

This study is limited in its findings, in that we only report on the outcomes of three workshops run in three different community groups. The findings would likely be very different were these workshops to be run again but with different communities. However,

this further serves to underline the thesis of this study, i.e. that qualitative research at the community level is an essential accompaniment to larger scale research projects that look at the way in which climate change is communicated. One-off workshops were used in this study, as we believe that it represents a model that could be most easily adopted by other researchers and for other communities. Additionally, this study was not designed to monitor the long-term impacts of these workshops; however, given the responses of the participants (and in particular the comments made by the Avonmouth group – see Section 3.1), such a study would likely yield interesting results. In addition to working with different communities and monitoring any long-term impacts, future studies could also adopt a similar approach to running workshops with several communities at a time. Furthermore, future workshops could also involve an element of reading and discussing poetry that had already been written (either by well-known poets, or by other communities in similar workshops) about issues that the community identified as being important, as doing so would allow participants to explore and discuss different perspectives and lifeworlds. As demonstrated in this study, the collaborative poetry writing worked well in allowing participants to explore each other's lived experiences in a creative and non-confrontational manner. Such an approach would also likely be successful in helping to bring together different (and perhaps opposed) communities by enabling them to discuss their lifeworlds in this way, as was exemplified by workshop involving the Manchester faith leaders (see Section 3.3).

Religion: Hearing how the participants from different faith communities engaged with the workshop was one of the most interesting parts of this article. I agree with the conclusion that working with faith leaders to develop dialogue across the diverse communities is a worthwhile initiative, and that awareness of different publics' perspectives, needs and worldviews is part of climate change communication. Therefore it strikes me that the article could do slightly more to engage with the relationship between religious discourses and ecological awareness in their own terms rather than too quickly putting them into an already familiar language of sustainability. The things in the discussion about neighbourly responsibility, or living in the moment with less focus on consumption, are not just connected to community experience but in part emerge from a religious world view that might complement but also find itself in tension with aspects of ecological communication. And while the implications need not be discussed in detail here they could perhaps be better acknowledged/signposted in a few sentences (e.g. in regard to Laudato Si, or even a Quaker sense of stewardship etc etc.).

We agree that the interactions between the different faith communities and their responses was very enlightening. We also agree that in presenting this discussion we should have at least acknowledged the potential of tensions between religious discourse and climate change communications, especially given the discussion that took place in terms of which which communities the faith leaders felt they did and did not belong to. As such, the following text has been added to Section 4 ('Discussion'):

The community of faith leaders had a similar outlook to the Stockport group, recognising that: "We must reduce the harm we cause / Both personal and corporate ware / A better carbon footprint / Before our world we tear." And that "To make the world more fair. / We need to change behaviour". As with the Avonmouth group, they also realised the need for education, and given their own positions within their communities they recognised that any initial activity likely needed to be driven by them. This was arguably a different type of individual responsibility than was evidenced in the other two workshops, as the faith leaders recognised that in some instances without their guidance and support for a particular topic action might not be instigated or even possible. In working with this community, it could be argued that effective climate change communications would provide reliable resources and frameworks for engagement that could then be shared by the individuals amongst their own communities and organisations. As was indicated by the participants themselves during this discussion, their sense of community is intertwined with their own religious worldviews, and as such several of these attitudes (e.g. 'overcoming prejudices' and 'addressing consumption') might be driven by religious practices rather than environmental concerns. It would also be interesting to further investigate what would happen if recommendations for

successful climate change mitigation strategies at the local community level clashed with the religious ideologies or discourses of a particular group. As Maxwell (2003, pp. 257) observed: "reductionist perceptions of reality are proving inadequate for addressing the complex, interconnected problems of the current age", and in addition to the benefits of working with such groups in tackling climate change, it would be worthwhile for future workshops to investigate the extent to which religious world views potentially clashed with climate change communications, and how different faith leaders reacted as a result.

References

FURMAN, R., RIDDOCH, R. & COLLINS, K. 2004. Poetry, Writing, and Community Practice. Human Service Education, 24. MAXWELL, T. P. 2003. Integral spirituality, deep science, and ecological awareness. Zygon®, 38, 257-276.